

# Reduction of windturbine generated seismic noise with structural measures

Rafael Abreu[1], Daniel Peter[2], and Christine Thomas[1]

[1]Institut für Geophysik, Westfälische Wilhelms-Universität Münster, Corrensstraße 24, D-48149 Münster, Germany
[2]Seismic Modeling and Inversion Group, King Abdullah University of Science and Technology, 23955 Thuwal, Saudi Arabia

**Correspondence:** Rafael Abreu (abreu@uni-muenster.de)

**Abstract.** Reducing wind turbine noise recorded at seismological stations promises to lower the conflict between renewable energy producers and seismologists. Seismic noise generated by the movement of wind turbines has been shown to travel large distances, affecting seismological stations used for seismic monitoring and/or the detection of seismic events. In this study, we use advanced 3D numerical techniques to study the possibility of using structural changes in the ground on the wave-path between the wind turbine and the seismic station in order to reduce or mitigate the noise generated by the wind turbine. Testing a range of structural changes around the foundation of the wind turbine, such as open and filled cavities, we show that we are able to considerably reduce the seismic noise recorded by placing empty circular trenches approx. 10 meters away from the wind turbines. We show the expected effects of filling the trenches with water. In addition, we study how relatively simple topographic elevations influence the propagation of the seismic energy generated by wind turbines and find that topography does help to reduce wind turbine induced seismic noise.

## 1 Introduction

The seismic energy generated by wind turbines (WTs) has been shown to propagate up to distances of 15 km and more (Schofield, 2001). This seismic energy or seismic noise can be measured by nearby seismic stations built for the detection of seismic events and/or seismic monitoring activities (Stammler and Ceranna, 2016; Neuffer and Kremers, 2017; Neuffer et al., 2019, 2021). The noise may result in the deterioration of the recording quality at seismic stations therefore leading to a conflict between seismological station owners and WT operators (Neuffer et al., 2019). However, since renewable energy is needed, we see an increase of the number of WTs around the world but the functionality and task fulfillment of seismic monitoring networks still has to be preserved (Neuffer and Kremers, 2017).

Most of the seismic waves generated by WTs that are influencing the seismic recordings are surface waves and especially Rayleigh waves (Gortsas et al., 2017; Neuffer and Kremers, 2017). The parameters of seismic noise produced (e.g., strength, frequency content) highly depends on the wind speed, height, number and type of the influencing WT (Neuffer and Kremers, 2017). The height of nearby WTs is affecting the frequency content of the noise wavefield in that ground vibrations generated by taller turbine towers are emitting lower frequencies, while smaller towers radiate higher frequencies (Neuffer and Kremers, 2017; Stammler and Ceranna, 2016). The frequency range of the WT induced seismic noise that affects seismic stations and



monitoring tasks lies in a range of 1–10 Hz (Hu et al., 2020; Zieger and Ritter, 2018; Friedrich et al., 2018; Marcillo and Carmichael, 2018; Stammler and Ceranna, 2016; Neuffer and Kremers, 2017; Neuffer et al., 2019; Zieger and Ritter, 2018).

Because the proposed distances between seismic monitoring stations and WTs of 15 km is not always fulfilled (Neuffer and Kremers, 2017), and often the distances are much smaller, solutions to these problems of WT noise interfering with seismic measurements still need to be found. A consensus between WT operators and seismological stations and seismic networks

is imperative for growth in the field of clean energy generation, and simple filtering operations to remove the seismic noise induced by WTs do not seem to be the solution to the problem (Neuffer and Kremers, 2017), however, advanced filtering methods may help to reduce WT noise and still allow seismic events to be detected.

A possible solution to this problem may be through the emerging field of seismic metamaterials. The original definition of seismic metamaterials are engineered media that acquire one (or more than one) property not found in naturally occurring

materials; these composites are usually designed using a combination of multiple elements arranged in repeating patterns, at one or multiple scales, that need to be smaller than the typical wavelength of the wave they aim to control (Brûlé et al., 2020). Following Brûlé et al. (2020) there are four main types of seismic metamaterials: i) seismic soil metamaterials, ii) buried mass resonators, iii) above surface resonators and iv) auxetic materials.

While most of these metamaterials are difficult to produce in large dimensions and since they are very expensive, their use

for mitigating the noise of WT is limited. However, in a recent study, the influence of tress on the seismic wavefield has been explored (Colombi et al., 2016b; Liu et al., 2019; Lim et al., 2021) and the presence of these trees been shown to lower seismic noise for a station places behind the trees. Buried mass resonators are, in principle, also useful candidates. However, they still possess very large dimensions and their construction is economically not feasible for attenuating WT noise. For instance, Palermo et al. (2016) have shown that in order to attenuate seismic waves for a frequency range of 1–10 Hz, one needs a

seismic barrier of buried resonators, each with heights larges than 1.5 m, radius of 0.5 m and weights around 6700 kg.

Seismic soil metamaterials may be a possible realistic candidate to mitigate the WT noise. Despite large dimensions, they are relatively cheap, because they may be constructed as an array of large holes with certain predefined shapes. Miniaci et al. (2016) has shown that one can mitigate seismic energy for a maximum frequency of 6 Hz with an array of cross-like cavities of 9 m wide by 10 m depth, separated by 2 m between them and arranged in an area of 100 m$^2$. A less restrictive experiment

has been carried out by Brûlé et al. (2017), where the authors show that they can mitigate the seismic energy for frequencies smaller than 10 Hz, by a grid of cylindrical holes properly distributed in the ground. These holes allow the distribution of the seismic energy inside the grid, producing an effect of dynamic anisotropy akin to an effective negative refraction index.

Based on the studies mentioned above, in this work we perform full 3D numerical wave propagation simulations that allow us to test the influence of structural changes such as cavities, trenches both filled and empty in order to reduce WT seismic

noise at seismological stations. We first start by modifying the numerical large-scale seismic soil metamaterials proposed by Miniaci et al. (2016) to understand the influence of the arrangement and number of unit cells that are necessary to obtain the desired attenuation results. Next we simplify the concept introduced by Miniaci et al. (2016) and Brûlé et al. (2017) and place simple circular holes (empty and filled with water) in front of the WTs and investigate how this configuration helps to mitigate the seismic energy. Continuing, we study how simple topographic elevations influence the propagation of the seismic energy





generated by WTs. We finally conclude the results of our investigations and propose the most appropriate scenario to avoid
seismic noise generated by WTs.

## 2    Numerical experiments

To mitigate the effect of WTs on seismological stations, we perform full 3D numerical simulations of elastic/acoustic wave
propagation using the freely available code specfem3d-cartesian (Komatitsch and Tromp, 1999). The code uses the spectral-
element method to solve the 3D elastic/acoustic equations of motion in the time domain. The use of full 3D waveform modeling
allows us to take into account the correct geometrical spreading of the seismic waves and to properly model surface waves. At
the boundaries of the domain, the code uses Clayton-Engquist-Stacey (Clayton and Engquist, 1977; Stacey, 1988) and/or PML
(Komatitsch and Martin, 2007; Komatitsch and Tromp, 2003) absorbing conditions to avoid unphysical reflections. For each
model we generate a complex hexahedral mesh using the softwares Trelis and MeshAssist (Gharti et al., 2017). Special attention
and effort is dedicated to the meshing process: it is a critical step in the modeling procedure since a good mesh guaranties the
good convergence of the numerical method. In particular, the spectral element method in combination with hexahedral meshes
leads to a symmetric mass matrix which allows to significantly reduce the computational cost of the numerical simulation while
keeping spectral accuracy of the solution (Komatitsch and Tromp, 1999). In the next sections we introduce different scenarios
to determine the most efficient way to mitigate the WT generated seismic noise.

### 2.1    Cross-shaped holes as metamaterials

First we consider the case of cross-shaped holes in the ground as presented by Miniaci et al. (2016) where these seismic soil
metamaterials were shown to attenuate the seismic wavefield sufficiently to protect buildings. Their cross-shaped unit cells had
the dimensions of $a = 10$ m, $b = 9$ m, $c = 2.5$ m and $H = 10$ m (see Fig. 1-e) and based on the Floquet-Bloch theory (Kittel
et al., 1996), the authors predict several frequency band-gaps between [2-6] Hz, a frequency range which is useful for our
purposes. However, the number and arrangement of individual unit cells needed to obtain the desired frequency band-gap is
not clear for seismological applications since Floquet-Bloch theory assumes periodicity in the structure (Gomez Garcia and
Fernández-Álvarez, 2015). To show the effect of these metamaterials on seismic waveforms, Miniaci et al. (2016) considered
an array of cross-like shaped unit cells distributed in a within a rectangular grid of dimensions $100 \times 100$ m$^2$.
        This kind of arrangement is too extreme for our purposes, however, it allows us to understand the effects of wave propagation
when we change the number of unit cells and their arrangement in order to keep the number of unit cells to the lowest possible
number, which ultimately will keep the cost and the total engineered area to a minimum. For this purpose we created twelve
different numerical models formed by different arrangements of individual cross-shaped unit cells (see Fig. 1-e). For each
model we consider an arrangement of 5×5 cross-shaped unit cells of dimensions presented in Fig. 1, covering five different
areas of dimensions $50(\times 50), 80(\times 80), 100(\times 100), 120(\times 120), 150(\times 150)$ m$^2$. For each of these models we also created
an additional model by shifting the intermediate layers of cross-shaped cavities (see Fig. 1–a and 1–b). Additionally, we
consider two more models where the distribution of cross-shaped metamaterial is circular (see Fig. 1–c and 2–d) to avoid





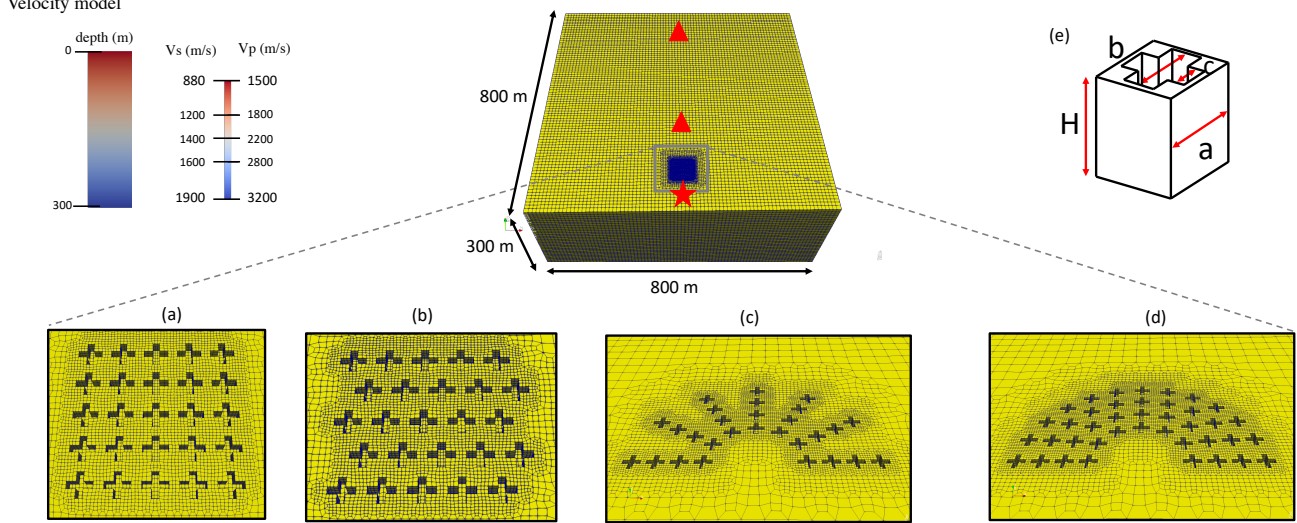

**Figure 1.** Mesh examples of the cross-shaped cavities used for the different numerical simulations. The red star indicates the place of the WT, the red triangles places of the seismic stations. Note that we considered seismic stations also towards the sides of the box. a) Grid of $5\times5$ cavities distributed in an area of $50\times50$ m$^2$. b) Same as a) but with two shifted lines of cavities. c) and d) cross-shaped cavities arranged in a half-circular arrangement. e) Unit cell detailing the dimensions of the cell, a, b and c and depth H.

diffraction around the structures and wavefront healing processes. The total dimensions of the models are $800 \times 800 \times 400$ m (length/width/depth). We numerically model a frequency range of seismic energy between [1-10] Hz with a Ricker wavelet centered at 5 Hz as a source time function. At the edges and bottom of the models we consider absorbing boundaries and at the top the free surface condition. Unlike previous studies (e.g. Miniaci et al., 2016; Palermo et al., 2016), the structural model is assumed to be a velocity increasing with depth, with varying velocities $v_p = 1500 - 3200\,m/s$ and $v_p = 1.7\,v_s$ and a constant density of $\rho = 2300\,kg/m^3$.

Results for the vertical (Z) component of seismometers located behind the metamaterials given in Fig. 1-a is presented in Fig. 2. We can observe that for the Ricker wavelet source with a dominant frequency of 5 Hz the seismic energy is not attenuated, on the contrary it is increased. This is likely due to interference of scattered waves from the different cross-shaped cavity walls. In addition, the waveforms change, also due to superposition of waves scattered from the cavity sides. Similar amplification results are obtained when shifting the individual cross-shaped unit cells (see Fig. 1-b,c and d). The shift of every second row with respect to the first seems to have little to no effect on the seismic waveforms, also for different distances. One needs to take into account that the wavelength of the propagated wavelet at the surface is about $1500(m/s)/5(\text{Hz}) = 300\,m$, almost half the total length of our models. Also, the location of the source is about 40 $m$ away form the first unit cell cavity. It thus seems that these kind of cross-shaped large-scale seismic metamaterials are not able to reduce seismic energy for our 5 Hz wavelet, but when we tested source wavelets with higher frequencies (15 to 25 Hz) the energy was attenuated. However, our





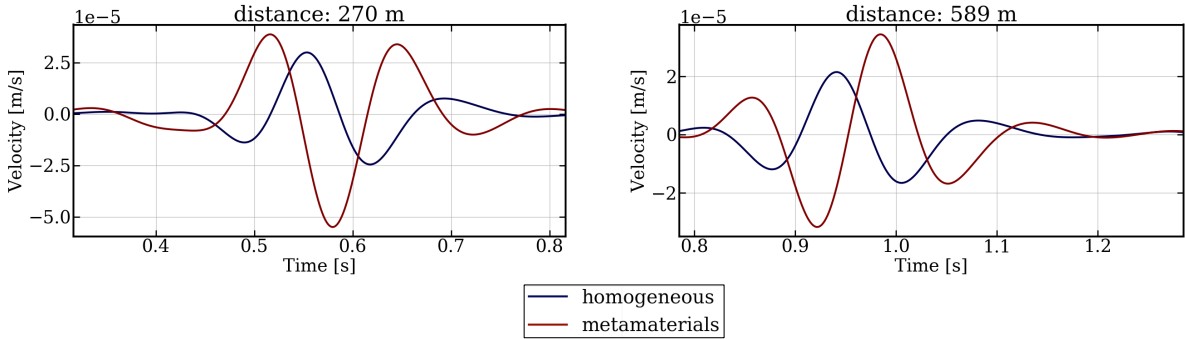

**Figure 2.** Simulation results for the cross shaped cavities (red line) in comparison with models without cavities (homogeneous model, blue line). The distance of the seismic station is indicated on top of each graph.

target frequencies for the attenuation of WT noise is in the range of [1–10] Hz, thus this size and type of metamaterial is not of practical use for our purposes, because they would have to have very large dimension for attenuating waves with frequencies
below 10 Hz thereby increasing costs and environmental impact.

## 2.2 Half circular trenches

We now consider simpler models compared to the cross-shaped metamaterials presented by Miniaci et al. (2016). To do so, we create a total of eighteen models with half circular trenches, nine of them empty and nine filled with water. We included varying depths of 20, 15 10 and 5 m and included two different widths of 3 and 5 m (see Fig. 3). Again we numerically model
a frequency range of seismic energy between [1-10] Hz with a Ricker wavelet centered at 5 Hz as a source time function. The point source is placed 10 m in front of the trenches while the stations are placed at a range of distances behind the trenches. We use a numerical model with dimensions of $400 \times 400 \times 200$ m (length/width/depth) discretized with more than one hundred million of global points (see Fig. 3). At the edges and bottom of the model we consider absorbing boundaries and at the top the free surface condition. The structural models are assumed to have constant velocities $v_p = 1500\,m/s$ and $v_s = 900\,m/s$
and density $\rho = 2300\,kg/m^3$. The reason for using constant velocities for this scenario is the fact that adding material to the trenches is computationally difficult to implement due to the creation of the meshes and we therefore resort to a simpler case for filled and empty trenches so that the difference in the seismic recordings is only due to the filling material for a better comparison.

Results for the vertical (Z) component seismic recordings for the model with empty trenches are presented in Fig. 4-a. We
can observe that all models attenuate the seismic energy in a similar way and only for 5 m deep trenches the attenuation is less pronounced. We also find that for all directions and distances of stations with respect to the WT, the model that best attenuates the seismic energy is a trench that is 5 m wide and 15 m deep. The deepest (20 m) and widest (5 m) trench shows effective



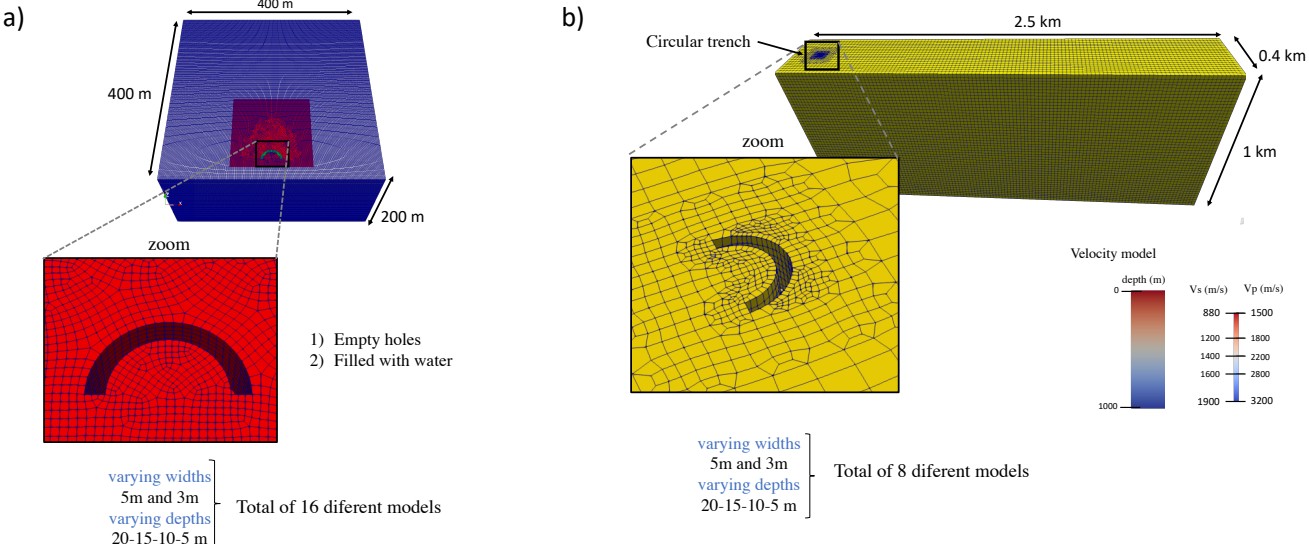

**Figure 3.** a) Mesh examples of models with half-circular holes either empty or filled with water with varying width and depths as indicated. For these models the velocities are constant. For more information see text. b) Mesh example of the large scale models created with empty holes with varying widths and depths. For these models, P- and S-velocity increase with depth as indicated. Seismic stations are placed across the entire surface 35 m apart.

attenuation results but it is not the best scenario. At larger distances (355 m) all models, excluding those with 5 m depth, behave virtually equal and at shorter distances (28 m) the best models are those with the deepest trenches.

Results for the models with trenches filled with water show a more complex behavior compared with empty trenches (see Fig. 4-b). This is because reverberations are generated by the presence of a fluid in the trenches. At short distances (28 m) a similar behavior is observed compared to empty trenches where the models with 5 m width and with 15 m, 20 m depth show the most attenuating effects and also at larger distances we can observe that some models still attenuate the energy similar to Fig. 4-a but the coda is longer than for the empty trenches due to the presence of reverberations in the water-filled trenches. At

the distance of 99 m, the 20 m deep and 3 m wide trench increases the seismic energy to higher amplitudes compared with the original seismic energy without any trench (purple line in Fig. 4-b). This indicates that filling the circular trenches with water, or indeed other material, may have the opposite effect to the desired attenuation of seismic energy, since amplification effects similar to those that occur in sedimentary basins can be expected (Olsen, 2000; Wirth et al., 2019). We tried models of trenches filled with other material, i.e., material with a different velocity and attenuation, however, the effect was the same as filling

them with water. Modeling porous small-scale material was not possible due to the size of possible meshes in combination with our frequencies and model sizes.

The results obtained in this section are, however, encouraging since we can observe a reduction of WT generated noise by placing half circular trenches between the WTs and seismic stations. These constructions lower the financial and environmental

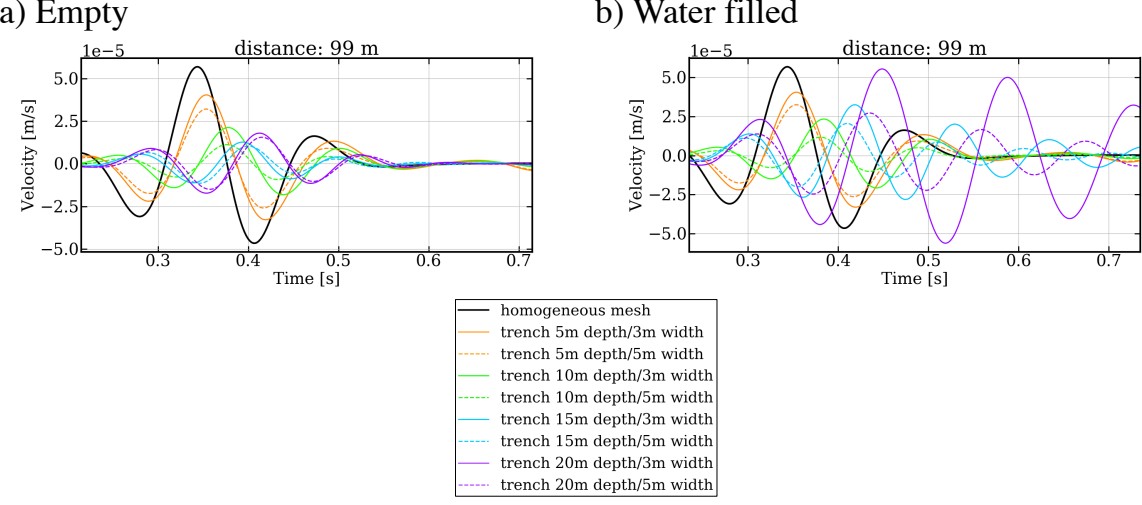

**Figure 4.** a) Simulation results for cavities as empty half-circular trenches (see Fig. 3) using a Ricker source time function centered at 5 Hz. Different sizes of cavities are shown by different colors (see legend) and the waveform of the model without cavity is shown as black solid line. b) Same as a) but for cavities filled with water.

impacts compared to results presented by Miniaci et al. (2016). Note that the above models were generated only for short distances between WT and stations, however, most seismic stations are more than 100 m away from WTs and we will explore a more realistic scenario in the next section.

## 2.3 Empty half circular trenches at larger distances

Encouraged by the results obtained in the previous section, we investigate how empty trenches can attenuate the seismic energy at large distances and in presence of structural changes in the soil (i.e., trenches) and with a more realistic sources. We create a total of eight modes with empty half-circular trenches within a model with dimensions of $2500 \times 400 \times 1000$ m (length/width/depth) discretized with more than one hundred million global points (see Fig. 3-b) with boundary conditions as above. The velocities in the model increase with depth as in the first scenario, with $v_p = 1200 - 3200 \, m/s$ and $v_s = 900 - 2400 \, m/s$ and a constant density of $\rho = 2300 \, kg/m^3$ (see Fig. 3 b) $-$ c)). Using this model allows to properly take into account the generation of surface waves at larger distances compared to the previous experiments, where we had to use a homogeneous velocity due to the complexity of the models with water-filled trenches.

Different to the experiments above, for this case we use source time functions that are taken from seismic noise measurements made by Neuffer (2020) and re-inject these at the place of the WT as a point sources for the three spatial coordinates. The seismic measurements by Neuffer (2020) were collected in the Windpark "Bürgerwindpark A31 Hohe Mark" located in Heiden (NRW, Germany) which consists of two WT concentration zones with three WTs per zone. Within the concentration zones,



the WTs are located about 500 meters apart and a nearby motorway is found in 500 m from the nearest WT. The identically
constructed WTs are of the type Enercon E-115. The WT with the largest distance to the motorway and to the other WTs
was selected as study object to conduct different measurements with 17 mobile seismic stations to identify the movements
of the tower, foundation and the immediately adjacent subsurface within the MISS Project (Minderung der Störwirkung von
Windenergieanlagen auf seismologische Stationen, Neuffer et al. (2021)). For our study, we use the seismic recording from
one accelerometer installed at a distance of 8 m from the WT. Following calculations made by Gortsas et al. (2017), we
select the magnitude of the point source to be of 78.202 MNm. Despite the assumed point source is too simplistic compared
to a realistic scenario were the WT type, aerodynamic conditions and foundations play a crucial role in the seismic noise
generation (Barthelmie and Pryor, 2006; Pryor et al., 2005; Barthelmie et al., 2006; Gortsas et al., 2017; Barthelmie et al.,
2007, 2010, 2016; Hu et al., 2018; Letson et al., 2019; Hu et al., 2020), it allows us to test whether empty half trenches can
attenuate complex waveforms within the frequency range of [1–10] Hz and with a realistic amplitude.

Results for the vertical (Z) component are presented in Fig. 5 as frequency spectra. Here we show spectra over waveforms
due to the complex nature of the source and to be able to detect whether any frequencies are attenuated or increased compared
with the model without structural changes (trenches) that is shown by the black line. In addition, previous studies also display
spectra rather than waveforms (e.g. Stammler and Ceranna, 2016; Neuffer and Kremers, 2017; Neuffer et al., 2019; Zieger
and Ritter, 2018) and we aim for a better comparison with those studies. In our results in Fig. 5, we can observe the overall
reduction of noise amplitudes for all frequencies when placing circular trenches between the WT and the seismic stations. The
models that most effectively reduce the seismic energy are those that are deepest (purple lines) with the wider trenches (dashed
lines) reducing the energy slightly better than narrower trenches (solid lines). Our half-circular trenches act as barrier to seismic
energy but for shallower trenches the energy of waveforms can still travel below the structure therefore the reduction of energy
is less pronounced here.

## 2.4 Topographic effects

As a last numerical experiment we change our model to include topographic variations at the surface. It is well known that
topographic variations have an effect on noise waveform amplitudes (Lacanna et al., 2014; Köhler et al., 2012) and it will be
instructive to see how WT noise is affected by simple topography since many WT are placed at the top of hills. We model this
scenario using the source measurements made by Neuffer (2020) as source input as described above. The model dimensions
are $2500 \times 1000 \times 1000$ m (length/width/depth) and we create topography in the shape of mounds with varying height of 33.5,
67, 100, 153 and 200 m (see Fig. 6-a). The velocity model for the bulk model domain (i.e., the box) is the same as above with
velocities increasing with depth. Inside the tomographic mounds we change the velocity, including higher and lower velocities
with and without random scattering media (see Fig. 6-b). All these models in Fig. 6-a and -b have the same topographic
horizontal extension and velocity variations, which guaranties that the differences observed in the simulations are only due to
the topographic elevations. The WTs are placed at the top of the mounds.

As mentioned before, in our numerical simulations, we consider that the topographic elevation may have a different velocity
perturbation compared with top layer of the bulk of the model domain, i.e., at zero elevation (see Fig. 6-g). This will introduce



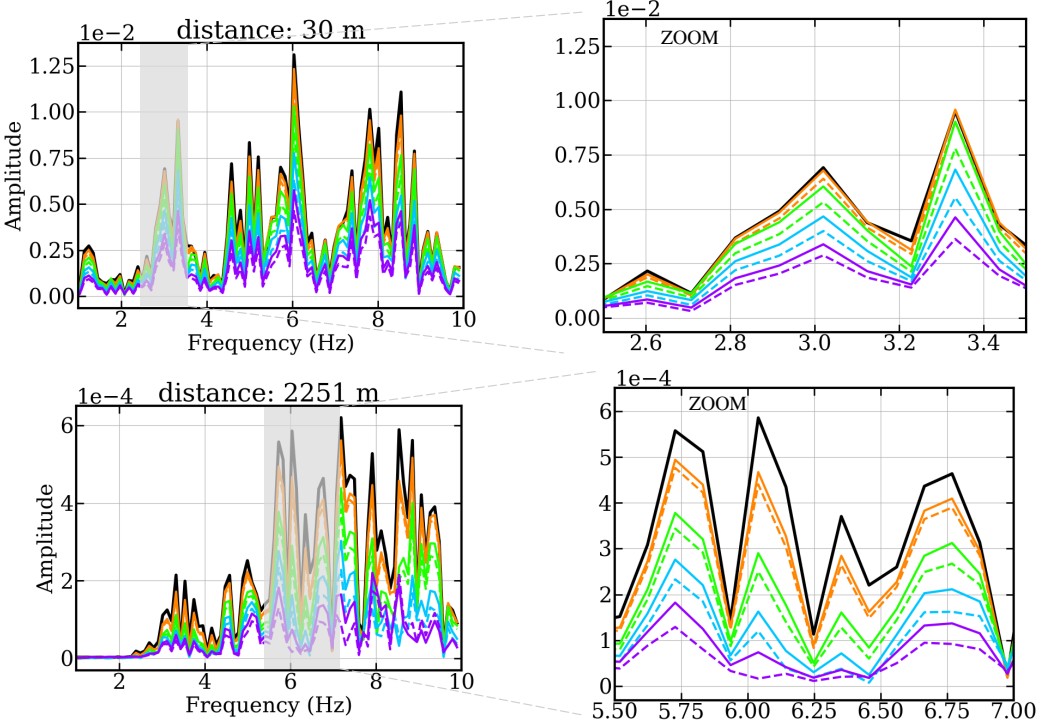

**Figure 5.** Frequency spectra of the simulation results for the seismic noise from the source time function from Neuffer et al. (2021) and in presence of half-circular trenches with varying dimensions compared with a model without trenches (black curve). See Fig. 3-b for the models considered here and Fig. 4 for the legend of models depicted by the colors.

an impedance (velocity x density) contrast at the bottom of the topography for the case of lower or higher velocities both with
and without scatterers. Therefore we expect changes in waveform and energy also due to these impedance contrasts.

Looking at different scenarios, we find that mounds with the same velocity as the top layer of the box reduces the recorded seismic energy for most frequencies for all topographic heights, and including scattering into these models emphasizes the effects. Higher mounds reduce the energy more efficiently than smaller mounds. If we use a velocity decrease inside the mound compared with the top layer of the box, we find instead increased energy for all frequencies and including scattering in
that model increases the energy even more. This can be explained in analogy to sedimentary basins where the trapped energy in the basin increases due to wave interference and depending on the structural geometry of the basin (Shumway, 1960; Olsen, 2000; Wirth et al., 2019). If, however, the velocity is faster in the mounds compared with the top layer of the box, the seismic energy recorded at the seismic station is reduced, and even further reduced is scattering is included (Fig. 7). As above, the reduction of the energy correlates with the height of the hills with larger hills reducing the energy more efficient. Because the
modeling of attenuation within the topographic region remains outside the capabilities of our numerical models, we instead included intrinsic attenuation in the entire numerical models and general observations remain virtually unchanged.

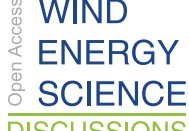

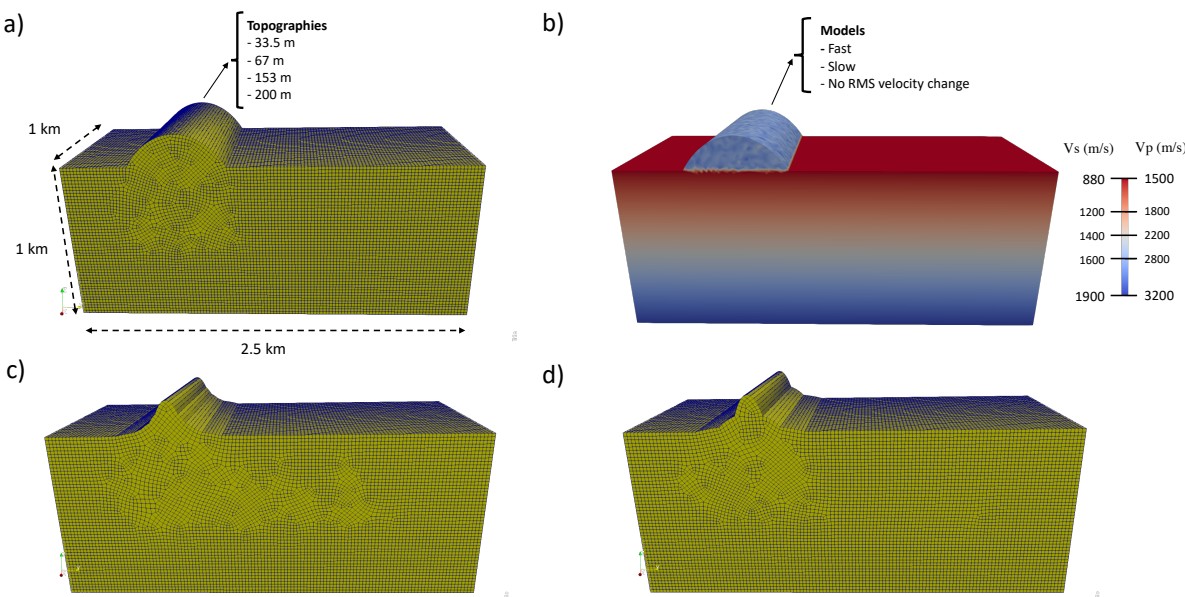

**Figure 6.** a) Mesh examples of large scale models with topographic mounds as shown. The length of the models is 2.5 km, topography heights are shown. b) Velocity model outside the mound with increasing velocity with depth as shown by the colors. Inside the mound, velocity variations with and without scattering are included as shown. With scattering the RMS velocity is either higher, lower or the same as the top layer of the box at zero elevation. Without scattering the velocity is is either higher, lower or the same as the top layer of the box. c) and d) two different models with more complex topography, c) smoother model, d) rougher model.

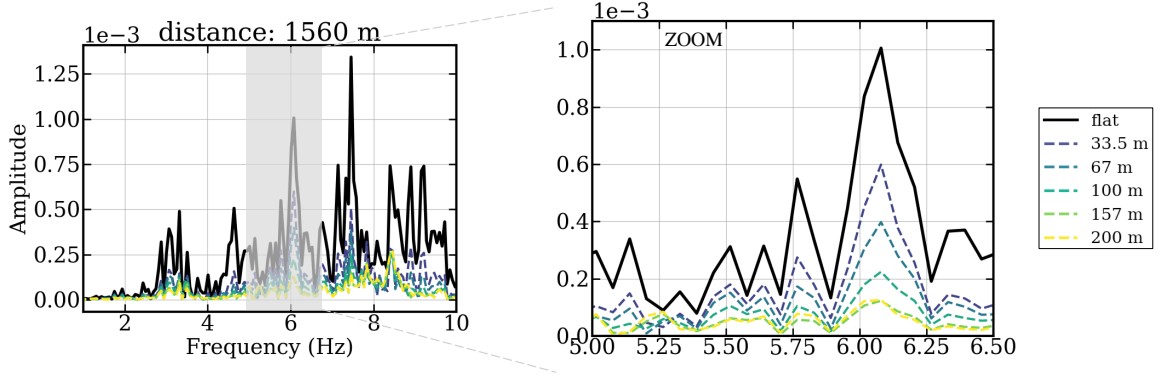

**Figure 7.** Spectra of the simulation results for the propagation of seismic energy in presence of topographic variations (see Fig. 6 for the models considered here). The source is placed at the top of the mounds, the mounds are filled with a scattering medium where the RMS velocity is faster than the top layer of the box (zero elevation).



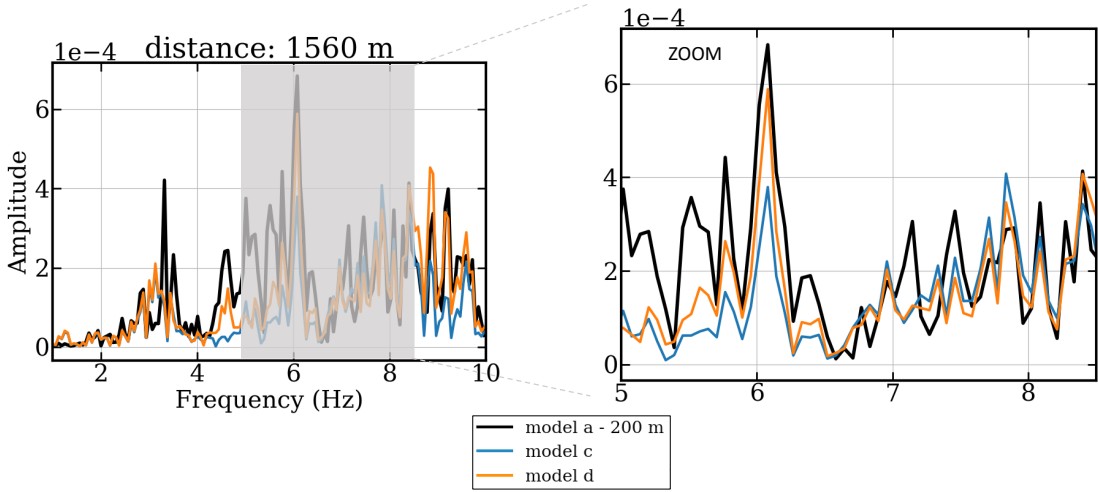

**Figure 8.** Spectra for the simulation results for the seismic noise for the models with complex topographic variations (orange and blue lines) in comparison with the 200 km high mound (black line). The source (WT) is placed at the top of the hill, the recording seismic station is placed 1560 km behind the WT. The models are presented in Fig. 6-a,c,d with 200 m height. We include scatters for which the RMS velocity variation is the same as the top layer of the box (zero elevation).

The mounds modeled here are very simple topography and one can expect that the amplification or reduction of the energy is dependent on the morphology of the topographic elevations. For evaluating how complex topographic variations affect the seismic noise recorded at stations behind the topographic variations, we consider two additional models given in Fig. 6 c-d.
Both scenarios variations have an elevation of 200 m and the topographic elevation has a random velocity perturbation of scatterers in a velocity model that is the same as the top layer of the box (i.e., at zero elevation). Results are presented in Fig. 8, where we compare to the simplified hill presented in Fig. 6 with the same height of 200 m as the top of the complex topography. We can observe that the complex topographies reduce the energy for some frequencies, for others increase the energy. This is true also for different distances of stations from the WT but it is not necessarily the same frequency for which the energy is
enhanced or reduced. We can observe that in general the amplitude/reduction of seismic energy will depend on the complex topography and will affect differently each particular frequency.

## 3 Discussion and conclusion

The demand of renewable energy systems increases every year around the world. In particular, the expansion of wind energy is expected to help renewable electricity generation to rise and it is expected to increase the most in absolute generation terms
among all renewables (Tabassum-Abbasi et al., 2014). This increase in the number of wind turbines conflicts with seismic stations since the noise generated by wind turbines is recorded at seismic stations (e.g. Neuffer et al., 2021, 2019; Neuffer and





Kremers, 2017; Stammler and Ceranna, 2016). Therefore it is imperative to find ways to mitigate the noise recorded at seismic stations in order to allow for the building of new WTs and contribute to the passage to renewable energy systems.

The mitigation of seismic noise is an active area of research today and the recent rise in the number of studies putting
solutions for seismic waves mitigation is large (Colombi et al., 2016a, b, 2020; Palermo et al., 2016; Zeighami et al., 2021).
Motivated by the study of Colombi et al. (2016a) we model different scenarios including structural changes on the wave path between the source of noise, i.e., the WT, and the seismic stations.

In the case of cross-like shaped cavities, we find no suitable attenuation, and instead the amplitude of the wave increased.
Contrary to Miniaci et al. (2016) the cross shaped cavities we used were too small to effectively attenuate the energy. Unlike
the case shown in Miniaci et al. (2016), where the cavities were closer connected to each other, here the energy still travel past the structural changes and amplifies through scattering effects and waveform interference.

To simplify the complexity of the cross-like shaped metamaterials by (Colombi et al., 2016a) and also potentially reduce dimensions and construction costs, we showed that we are able to effectively mitigate WT noise within the frequency range of [1–10] Hz with half-circular trenches at 10 m distance from the WTs between the WT and the seismic stations. This reduction
is seen for distances of 2.5 km and therefore we conclude that this scenario is a possibility to mitigate the effects of WT noises on seismic stations. However, the filling of the trenches has the opposite effect, due to reverberations of energy within the trench, if it is filled with water or other material. Therefore the trenches, if empty, act like a barrier to seismic energy, and in order to reduce energy efficiently, they need to be deep enough so that the energy cannot diffract around the bottom of the trench. The fact that filling the trenches with water or other material may oppose the desired effects is important to
take into consideration because for realistic soil environments, the integrity of the trench can be compromised by having, for example, non consolidated sediments. Alternative solutions to this situation can be keeping the integrity of the trench with a cement casing as done in the oil industry (Davies et al., 2014). These results are consistent when considering a Ricker source or injecting seismic noise generated by WTs (Neuffer et al., 2019) and using a realistic magnitude (Gortsas et al., 2017). Despite the measurement used as source of seismic noise belong to a single experiment made by Neuffer et al. (2019), our results
should be consistent when considering different sources of WT noise since the energy reduction is observed within a complete frequency window of [1–10] Hz.

Our numerical simulations of WT noise propagation in presence of topography show that terrains with topographic elevations can help to mitigate the seismic noise recorded at seismological stations, however, modeling a mound with low velocity material, also with scattering instead increases the energy recorded at the seismic stations. This in is contrast to the case of the
"Energieberg" (hill for energy production) in the center of the city of Karlsruhe, Germany. At the top of the hill, three WTs and a photovoltaic system are installed. This hill is around 60 m high and is a disposal site for waste, which seems to produce a strong damping of seismic signals. Zieger (2019); Ritter (2020) conducted several seismic measurements WTs placed at a distance to the hill, in order to determine the influence of the subsurface on WT-induced seismic signals for this special case. They found that the WT induced seismic signals are not visible at distances of 130 m (Zieger, 2019), making this hill a form
of metamaterial.

The decline of the seismic amplitudes along the measuring profile away from the hill may be explained by an impedance contrast at the bottom of the waste disposal site between the highly unconsolidated waste material and the natural sediments of the Upper Rhine Graben (Zieger, 2019). Our models include such an impedance contrast at the bottom of our imposed topography and for low velocities we measure energy increases, therefore we assume that especially the attenuation of uncon-
solidated waste inside the hill is responsible for the seismic noise reduction. With our numerical models we cannot include such attenuation effects. But previous studies showed that unconsolidated material filled with cracks or poroelastic materials generate different attenuation effects leading to reduction of the seismic energy (Zieger, 2019) or an increase (Hunziker et al., 2018; Müller et al., 2010; Johnston et al., 1979; Toksöz et al., 1979; Biryukov et al., 2016).

Numerical simulations combining different soil parameters such as porosity and plasticity have not been considered in this
study due to numerical capabilities limitations, however their role may be crucial to design the best scenario to attenuate seismic noise emerged from WTs (e.g. Ghaedizadeh et al., 2016; Bessa et al., 2019; Meng et al., 2021; Ji et al., 2020; Mirzaali et al., 2017; Amireddy et al., 2018; Wang et al., 2019). New generations of numerical codes with the necessary capabilities including these effects (e.g. Colombi et al., 2020) will allow a more realistic design of scenarios that will help to mitigate the WT generated seismic noise.

*Code availability.* The specfem3d-cartesian code is freely available through the webpage of the Computational Infrastructure for Geodynamics (CIG).

*Author contributions.* C.T. designed and directed the project. R.A. and D.P. performed the numerical experiments. R.A. and C.T wrote the manuscript.

*Competing interests.* The authors declare that they have no conflict of interest.

*Financial support.* R.A. and C.T. acknowledge funding from the MISS research project funded by the "Europäischer Fonds für regionale Entwicklung (EFRE)".

*Acknowledgements.* R.A. acknowledge help and continuous support from Stefan Klingen and Christian Maas for installing and running specfem3d-cartesian at the cluster of the University of Münster as well as the help of Hom-Nath Gharti in the process of the mesh creation. The seismic data used as source input were provided by Tobias Neuffer.



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
