# Peer review of "Reduction of windturbine generated seismic noise with structural measures"

_Wind Energy Science, 2022_

## Community Comment (CC1)

This paper carries out different numerical simulations to investigate the influence of different ground/soil structural changes (so called the seismic metamaterials according to this paper) around a wind turbine on the seismic wave-path between the wind turbine and the seismic station. It's a very interesting and meaningful research in the reviewer's opinion. The authors investigate four main types of different metamaterial scenarios including cross-shaped holes, half circular trenches at short distances, empty half circular trenches at large distances, and different topographic effects. The research is new, the discussion is insightful, and the English writing is clear and straightforward.

We would like to thank the reviewer their nice words and for their helpful comments. We provide clarifications below.

Three minor thoughts the reviewer would like to share:

1. It seems this paper only studies the seismic signal emitted from one wind turbine. Will the investigated structural changes have similar influence on the seismic wave-path due to multiple wind turbines (or a wind farm)? It will probably make the research much more complex, but it may be worth discussing more, especially for the case of empty half circular trenches at large distances. Usually there is a wind farm at a large distance from a seismic station (or seismic stations).

R: In principle, it is expected that we will observe similar results for multiple WTs. The idea is to place the circular trench at each WT and in direction of the seismic station. The question, however, is how two wind turbines with two trenches would reduce the noise.
To answer this, we have created a new scenario where two WTs are placed separated by a distance of 200m. In front of each WT we place a circular trench of 5 x 15 m (width-depth). The numerical model has a total dimensions of 2.5 x 0.8 x 1 km (length-width-depth) and it has an increasing velocity with depth, just like in the previous scenarios. Results in the Fig. below show that we are still able to efficiently attenuate the seismic energy at large distance. We have added a paragraph in the main text.

2. If we think in a different way, will the structural changes help mitigate noise if these structural changes are close to the seismic station but not the wind turbine.

R: Seismological stations are often placed in strategic locations in order to do seismic monitoring of , e.g. a dam or similar structures, or to record earthquake signals comming from all around the world and many other activities. Placing the structural changes close to the seismic stations would then filter the WT turbine noise but it will also filter the rest of the seismic signals the seismic station is supposed to record. Therefore, the trench must be close to the wind turbine.

3. Is it safe for a wind turbine if digging a 5-20m depth trench which is only 10 m distance from the wind turbine? Well, this may be beyond the scope of this research. It is only the reviewer's curiosity.

R: The distance between the circular trench and the WT will depend on many technical details. In our study we aimed to place the WT close to the trench in order to observe the effects. We also tried scenarios with larger structures and different distances. When, for technical reasons this distance is not safe and/or cannot be fulfilled for any other reason, then it is possible to place the trenches in other locations, purely from a noise reduction point of view. For this we would then have to carry out more numerical experiments. Question about the stability of a windturbine would have to be tested by the engineers.

Some technical corrections:

1. It may need to change the "windturbine" to "wind turbine" in the title;

2. Page 3, line 83, "distributed in a within" should be "distributed within";

3. Page 7, line 157, "of the WT as a point sources" should be "of the WT as a point source"?

4. In the fourth line of the caption of Figure 6, there are two "is".

5. Page 12, line 249, "This in is contrast" should be "This is in contrast".

R: Than you, we have corrected them.

[Figure]

Figure: a) Setup with two wind turbines, each with a trench in the direction of the seismic station. b) The spectra of the model without trenches (dashed lines) and with trenches (solid lines). The orange/red lines indicate the case of two wind turbines, the grey/black lines those of one wind turbine.

---

## Author Comment (AC2)

General remarks

The paper is well written and easy to follow, therefore in my opinion there is no need for major changes in the text. The authors theoretically evaluate possibilities to reduce the influence of wind turbines (WT) in seismic records by placing empty or filled cavities into the travel path of seismic noise waves. The efficiency of the noise reduction depends on the shape of the cavities and on the observed frequencies. When focusing on the relevant frequencies between 1 and 10 Hz which are observed in many previously published examples at distances above 1 km from the WT it turns out that the most effective noise suppression (among the investigated models) is achieved with empty half-circular trenches of certain dimensions. Regarding the ongoing discussions on the necessity of restrictions of WT operations in the vicinity of seismic stations this is a valuable contribution to the search for remedies in this conflict of interests between the operators of seismic stations and the operators of WT. It shows that there exists at least a theoretical possibility of noise suppression, however, the practical feasibility remains unclear and this probably cannot easily be answered by seismologists. Below I collected some questions concerning this topic. Details need to be further investigated also by engineers and constructors of WT. Some part of the paper deals with the influence of topography on seismic wave propagation. The result of this section, however, does not lead to unique recommendations concerning the possible locations of WT if the geologic structure of the whole area is not really well known. So there might be noise reductions or enhancements depending on the velocity relations between the mound-like structures and the ground below. Also the frequencies affected depend on the dimensions and shape of the hill structures and are not easily predicable. Only in specific cases the location of a WT on top of a hill does help to reduce emitted noise. It is good to know that topography may influence the propagation of noise but the mentioned uncertainties and the fact that in many cases the location of WT is restricted by a number of site selection conditions will most probably not make topography a decisive factor in noise mitigation. To give an example, one could have a look at the station GR.GRB1 (https://opentopomap.org/#marker=13/49.3903/11.6506) where 5 WT are installed in an approximate distance of 3 km to the north-east of the station (marked in the opentopo map). The WT are located on top of a small mountain range with an elevation difference of about 100m to the station level. The noise spectrum of GR.GRB1 shows one of the largest WT noise peaks at 1.1 Hz observable at German permanent stations:

https://www.szgrf.bgr.de/cgi-bin/send_windspec.py?station1=GRB1&station2=None&year1=2017&year2=None&night1=Night&night2=Night&minfrq=0.9&maxfrq=8.0&lopsd=10.0&hipsd=1.0e5&linlog=linlog&smooth=3&operation=overlay&shownumbers=nonumbers&submit=Create+Figure

Of course, this does not prove anything, but it is an example that even with WT installed on top of a hill there may exist very large noise signals.

We would like to thank the reviewer for their nice words and for their helpful comments. We provide clarifications below.

Specific comments/questions

What is the radius of the half-circle trenches? This is nowhere explicitly mentioned. The reader just can guess from the "distance of the WT from the trench" that it is 10m.

R: The radius is 10m. We locate the WT inside the circular trench in order to avoid lateral noise being propagated. We have added this to the manuscript.

---> As far as I know the concrete foundation of a WT can have dimensions (diameter) in the order of 20m. That means that the trench is right at the border of the foundation (if not within). I guess this cannot be realized as it affects the stability of the WT and therefore, I see problems to install circular trenches in proposed dimensions. What radius would be necessary in practice instead? The trench should be significantly decoupled from the foundation otherwise there will be no mitigation effect.

R: The main motivation of our simulations was not to design of circular trenches with proper dimensions to be installed in the field, but to show that the idea works and seismic noise can be attenuated. Our simulations show that the seismic noise can be attenuated when installing an empty trench at the border of the WT. Our WT in the simulations has no foundation.

To answer the question of what radius will be necessary instead, we would have to run simulations including foundations and a number of circular or half-circular trenches, which is beyond the scope of this study. Our aim was to show that the trenches work in reducing the seismic wavefield. Geology and topography may also play a role.

---> I understand that using a point source makes things easier and when looking from large distances principal effects may be well described by such a simplification. However, the foundation of a WT plays a major role in the generation of seismic waves by its movement as a whole in different directions. If the source dimension and the dimension of the wave blocking cavity around it are in the same order, how much can we rely on the results using a point source approximation?

R: We are not trying to design the realistic dimensions of the half circular trench; however, the reviewer raises an interesting question about the dimension of the foundation of the WT with respect to the empty trench. This should be investigated in future simulations, including realistic setups. However, some preliminary tests show that if the trench is located further away from the sources (i.e. 5 point sources spread over the width of the foundation region), the difference is that the trench needs to be deeper but it still reduces the signal. Unfortunately, running more simulations is not possible in the short revision time but several months would be needed to properly investigate these effects.

---> Can you descriptively quantify the mitigation effect somehow? For example, how many more WT with trenches could be installed in comparison to one without to have the same effect at the station? Or how closer a WT with trench could be situated to a station compared to one without?

R: These are questions that we believe cannot be answered easily. They depend on too many variables like topography, geological materials and structures located between the WT and seismic stations, etc. In the hypothetical case that we are able to predict these kinds of effects for one specific case scenario, they likely cannot be generalized.  In future work we will consider testing different scenarios.

---> Since water in the trenches seems to foil the mitigation effect, the trenches need to be properly protected against intrusion of water (from below and above). On the other hand the two walls of the trench should be decoupled from each other as much as possible to prevent vibrating of the trench stabilizing construction as a whole. This seems to be a quite challenging task for the constructors of such an installation.

R: Yes indeed, the efficient decoupling of the wall is a problem to solve. This can be done, for example, using springs or high attenuative materials. In fact, this idea directly leads to the concept of auxetic metamaterials placed between the walls. One could, for example, design inside the walls certain auxetic metamaterials with well known (predicted) properties that will trap seismic waves in certain frequency range. This will stabilize the walls and trap the energy, but of course, it needs further studies also in the field of engineering.

---> I could imagine that readers would be interested in the computing costs of your simulations. It is of course strongly dependent on the hardware, but giving an idea on the computing times would be helpful.

R: We run each numerical simulation on 10 cores with 720 processors in total. The total simulation time is about 2 hours. WE added this to the text.

---> Technical corrections (giving line numbers):

25: affected range 1-10Hz, correct for distances above 1km or so. At smaller distances higher frequencies will be observed.

40: tress -> trees

45: larges -> larger

48 *have* shown

114: comma missing

116: "placed 10m in front of the trench": is this the center of the (half-)circle?

152: "as in the first scenario" -> "as in the scenario with cross shaped holes"

196: reduces -> reduce

204: efficient*ly*

figure 3a) and 3b)  diferent -> different (2 times)

R: Thank you, we have corrected them.